# Network vulnerability of cattle movement in Minas Gerais, Brazil, from 2013 to 2022

Anna Cecília Trolesi Reis Borges Costa[ID][1,☯], Lara Savini[2,☯], Luciana Faria de Oliveira[3], Andrey Pereira Lage[4], Elaine Maria Seles Dorneles[ID][1]*, Luca Candeloro[2]

1 Departamento de Medicina Veterinária, Escola de Zootecnia e Medicina Veterinária, Universidade Federal de Lavras, Lavras, Minas Gerais, Brazil, 2 Statistic and GIS sector, Epidemiology Departament, Istituto Zooprofilattico Sperimentale dell'Abruzzo e del Molise "Giuseppe Caporale", Teramo, Italy, 3 Instituto Mineiro de Agropecuária, Belo Horizonte, Minas Gerais, Brazil, 4 Departamento de Medicina Veterinária Preventiva, Escola de Veterinária, Universidade Federal de Minas Gerais, Belo Horizonte, Minas Gerais, Brazil

☯ These authors also contributed equally to this work.
* elaine.dorneles@ufla.br

## Abstract

The analysis of networks from cattle movements is an important approach to investigate areas and premises where disease outbreaks can occur and be contained. Vulnerability analysis allows a more profound understanding of the network by combining network measures with the strategic removal of nodes, helping to identify more vulnerable areas and the best metrics to support disease control planning. Therewith, the aim of this study was to analyze the network vulnerability of cattle movements from 2013 to 2022, in Minas Gerais, Brazil and to identify the spatial spreaders into the network to improve infectious disease control programs by targeted risk-based surveillance and intervention. The vulnerability was calculated considering the graphs diameter and the spatial spreaders with a threshold distance of 300 km, for incoming (IN) and outgoing (OUT) movements. Additionally, a risk-based analysis was performed in the more vulnerable region. The results showed Triângulo Mineiro/ Alto Paranaíba with higher vulnerability and many IN spatial spreaders, as well as Vale do Mucurí region with many OUT spatial spreaders. The risk-based analysis revealed betweenness and out degree as the most effective measures to be considered for intervention. Therefore, the vulnerability analysis and the spatial spreader were observed as great tools for risk-based interventions and surveillance. Furthermore, Triângulo Mineiro/ Alto Paranaíba and Vale do Mucuri regions were important regions, considering restriction of animal infectious disease spread in Minas Gerais, Brazil.

**Data availability statement:** The data underlying the results presented in the study are available with Instituto Mineiro de Agropecuária, https://www.ima.mg.gov.br/, contact: ima@ima.mg.gov.br The data used for this project contain identification details of the premises such as name, latitude and longitude, used in the analysis, the availability of these information would have to be authorized by the owners and that is not possible to acquire in light of the great number of farms in Minas Gerais state, Brazil. Additionally, Brazil has a law Lei Geral de Proteção de Dados Pessoais (LGPD), Lei n° 13.709/2018, that prohibit us to share or even access of some the personal information that was in our data.

**Funding:** The authors would like to thank the Coordenação de Aperfeiçoamento de Pessoal de Nível Superior, Brasil (Capes), Fundação de Amparo à Pesquisa do Estado de Minas Gerais (Fapemig) (RED 000132-22) and Conselho Nacional de Desenvolvimento Científico e Tecnológico (CNPq) (402774/2022-1), Brazil, for the financial support. EMSD is thankful to CNPq for her fellowship. The funders had no role in study design, data collection and analysis, decision to publish, or preparation of the manuscript.

**Competing interests:** The authors have declared that no competing interests exist.

## Introduction

Cattle movements are often the primary pathway for spreading infectious diseases in a population. Therefore, understanding cattle movement patterns is crucial for assessing the transmission risks of highly infectious diseases during epidemics. In this scenario, considering that animal movements constitute complex systems, network analysis can effectively untangle them to help in infectious disease control and prevention [1–5]. Indeed, the network analysis is an approach that elucidates the relationships among premises and the implications of those relationships in cattle infectious disease transmission [1]. In recent years, this technique has emerged as a valuable tool for understanding disease dynamics, designing surveillance strategies, and implementing targeted interventions [2–4]. This approach enables the identification of network metrics that highlight key nodes within the system structure which should be the focus of intervention [5].

In fact, once the network is described, the identification of nodes to be intervened, to avoid disease spread, is usually based on the network metrics [6–8]. However, the graph can also be analyzed regarding its vulnerability, a function that measures how vulnerable a network is based on the number of nodes reached starting from random seeds (premises), similarly to the diffusion models based applied to networks in other contexts [8,9]. The vulnerability function accounts for connections and the diameter of the network, being more vulnerable the network that is more connected [10]. Additionally, vulnerability of a graph can also indicate the best measures to be considered for interventions inside the graph system, identifying the best nodes to be blocked in case of disease spread over the network [8–10], allowing for focused implementation of human and financial resources in animal disease surveillance and control programs.

Furthermore, animal disease control programs can also be improved with the identification of spatial spreaders (or hubs), which are the super spreaders and super susceptible nodes [11,12]. Super spreaders are nodes with the potential to spread diseases over very long distances and to many other nodes, and the super susceptible are places (premises) with a high probability of acquiring diseases since they receive movements from many other nodes [13]. The identification of the spatial spreaders allows for intervention in these nodes, therefore the spread of an infectious disease would be contained into a small area, increasing the efficacy of control measures and preventing diseases from disseminating far away through the network [13].

The network analysis of cattle movement in Minas Gerais was shown elsewhere [14], describing a very connected network with movements more focused in the west and east sides of the state, being the Triângulo Mineiro/ Alto do Paranaíba and Vale do Mucuri regions of great emphasis on movements and cattle population [15]

In this sense, the aim of our study was to analyze the network vulnerability of cattle movements in Minas Gerais, Brazil, from 2013 to 2022 and to identify the spatial spreaders into the network to improve animal disease control programs by targeted risk-based surveillance and intervention.

## Materials and methods

### Study location

Minas Gerais state is in the southeast region of Brazil, at latitudes 14°13'58" and 22°54'00" south and longitudes 39°51'32" and 51°02'35" west, divided into 853 municipalities, grouped into twelve regions: Northwest Minas, North Minas, Jequitinhonha, Vale do Mucuri, Triângulo Mineiro/Alto Paranaíba, the Central Minas, the metropolitan area of Belo Horizonte, Vale do Rio Doce, West Minas, South/Southeast Minas, Campo das Vertentes and Zona da Mata, according to Instituto Brasileiro de Geografia e Estatística (IBGE) in 2022 (https://www.ibge.gov.br/geociencias/organizacao-do-territo-rio/estrutura-territorial/23701-divisao-territorial-brasileira.html) (Fig 1). The state climate is classified as Aw (tropical savannah climate with dry winter season), Cwa (humid temperate climate with dry winter and hot summer), and Cwb (humid temperate climate with dry winter and moderately hot summer) [16]. The state covers an area of 586,513,983 km², with a population of 20,538,718 people in 2022 [17] and 22,993,105 cattle heads in 2022 [18]. To provide a clear geographical context for the analysis, Fig 1 illustrates the twelve regions of Minas Gerais, each marked with its respective number (01–12).

Data on cattle movements in Minas Gerais state, Brazil, from January 2013 to December 2022, were obtained from the Animal Transit Guide (GTA – *Guia de Trânsito Animal*), provided by the Instituto Mineiro de Agropecuária (IMA), the official animal health authority of Minas Gerais, Brazil. The datasets were organized following the same pattern (same number and names of variables) in all years. Therefore, all variables were reorganized, or excluded, in case it was not necessary for the network analysis (detailed ahead), so every yearly dataset presented the same format and structure. Additionally, when necessary, the local identification code, for origin and destination, was standardized for analysis. Therewith, the final GTA databases were structured according to the description performed elsewhere [14]. For all analysis, movements

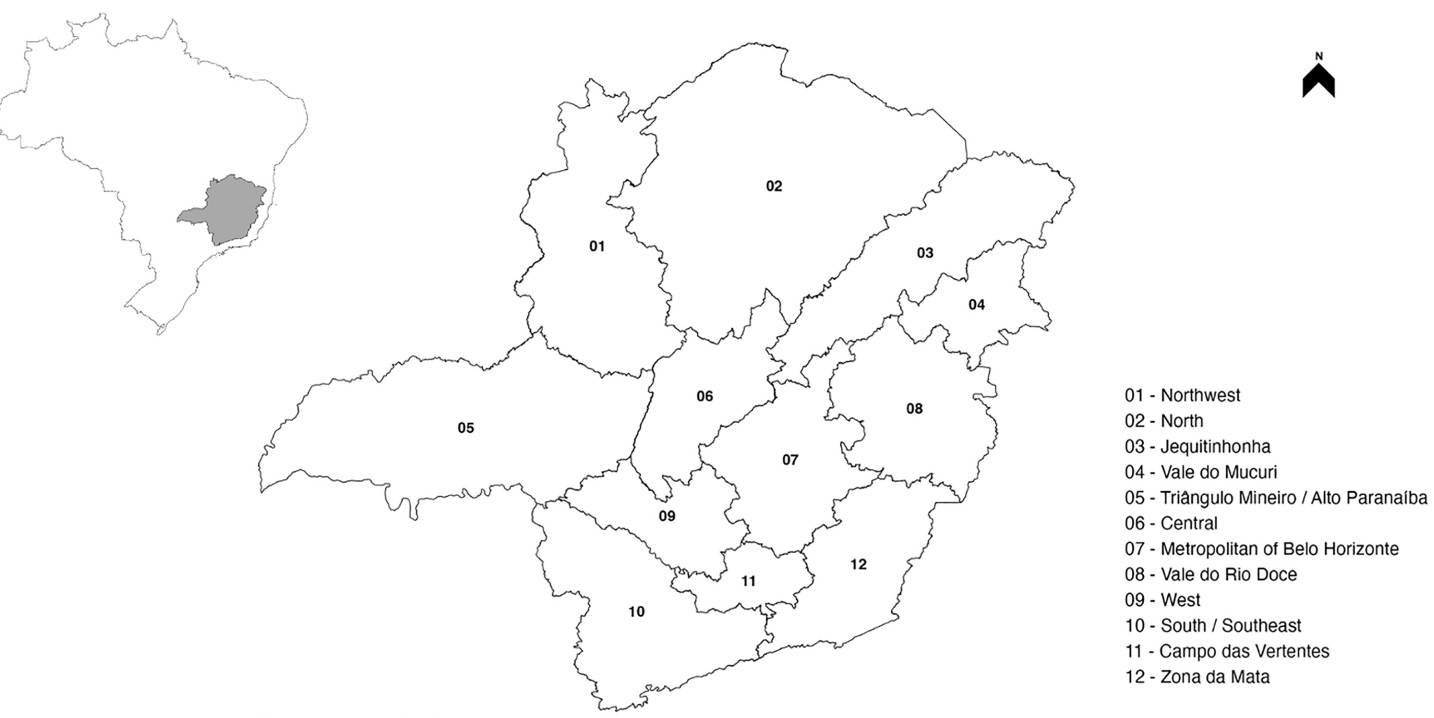

01 - Northwest
02 - North
03 - Jequitinhonha
04 - Vale do Mucuri
05 - Triângulo Mineiro / Alto Paranaíba
06 - Central
07 - Metropolitan of Belo Horizonte
08 - Vale do Rio Doce
09 - West
10 - South / Southeast
11 - Campo das Vertentes
12 - Zona da Mata

**Fig 1. Minas Gerais with numbered regions from 01 to 12 and a scale in kilometers.** Data source and description.

to slaughterhouses were removed. The georeferenced premises were checked to ensure their geolocation was correct inside the Minas Gerais state, and incorrect ones were eliminated [19–23] The datasets were grouped by local of origin, local of destination, region (origin and destination) and month per year to create a list of the network objects, from which the vulnerability R function (S1 Function) was applied.

## Definitions of network metrics used in this study

To clarify the key network measures used in this study and their relevance in understanding the dynamics of disease spread, a glossary is provided, which includes definitions of each measure, their network meaning disease implications, and the nodes and links/edges used in the analysis (Table 1) [24–28].

## Local regions vulnerability analysis

Network vulnerability was defined as the structural potential of a network to propagate an infection starting from a limited number of initially affected nodes, based solely on its topology. This metric estimates how many nodes can be reached through directed links when a given proportion of premises is considered infected.

**Table 1. Glossary of key network measures, their level of use and their importance in understanding disease spread dynamics.**

| Measure | Level | Definition | Application to disease relevance | Reference |
|---|---|---|---|---|
| Average path length | Network | The average length of the shortest paths between all pairs of nodes in the network. | A shorter average path length indicates a higher potential for rapid disease transmission across the network. | Hearst et al., 2023 |
| Betweenness | Node | It measures how much a node acts as a bridge or "bottleneck" within a network by lying on the shortest paths between other pairs of nodes. | Highlights key locations acting as bridges in the network, crucial for controlling disease spread across regions. | Hearst et al., 2024 |
| Big Cliques | Network | A large subgraph where every node is connected to every other node, forming a network of overlapping triangles. | A large group of nodes where each node is interconnected, forming extensive pathways that facilitate rapid disease spread. | Eppstein et al., 2010 |
| Cliques | Network | A subgraph where every node is fully connected to every other node. The simplest form is a triangle of three interconnected nodes. | The idea of connectivity among the nodes, enabling rapid disease to spread due to the high level of connectivity among nodes. | Bryś & Lonc, 1998) |
| Diameter | Network | The longest shortest path between any two nodes in the network. | Determines the maximum distance disease might need to travel between the most distant nodes, impacting the speed of spread. | Cardenas et al., 2021) |
| Edges Density | Network | The ratio of actual edges to the total possible edges in the network. | Indicates the overall connectivity of the network, with higher density suggesting a higher risk of widespread disease transmission. | Hearst et al., 2023 |
| Giant strongly connected component (GSCC) | Network | The largest subgraph of a directed network, where there is a path from each node to every other node within this subgraph. | If a pathogen enters the GSCC, it can easily be transmitted throughout all the nodes in this component. | Kolaczyk and Csárdi 2020) |
| Indegree | Node | The number of incoming connections that a node has in a directed network | Nodes with high indegree are very susceptible to became infected from multiple other nodes in the network. | Hearst et al., 2023 |
| Outdegree | Node | The number of outgoing connections that a node has in a directed network | Nodes with high outdegree are great spreaders of disease to multiple other nodes in the network. | Hearst et al., 2024 |
| Transitivity | Network | The ratio of the number of closed triplets (triangles) to the number of connected triplets of nodes. | Higher values of transitivity suggest easier disease transmission among the nodes composing the clusters within the network. | Luke, 2015 |
| Unreachability Ratio (UR) | Network | The ratio of node pairs in a directed graph that cannot reach each other, relative to the total number of possible pairs. It is calculated as the number of pairs with infinite distance divided by the total number of node pairs. | A high UR indicates a network with lower connectivity, suggesting reduced potential for disease spread among nodes. | Kumar & Helmy, 2010) |

To calculate network vulnerability, we used a custom R function (Vulnerability.table) developed with the "igraph" and "tidyverse" packages. The function simulates the reachability of infected nodes by randomly selecting different percentages of seed nodes (ranging from 1% to 30% of all nodes) and measuring the number of unique nodes reachable from them through directed paths. For each seed percentage, the function performs 1,000 simulations and returns the mean and standard deviation of the proportion of nodes reached. Reachability is calculated using ego-networks of radius equal to the network diameter, in "out" mode (i.e., considering all downstream connections). The vulnerability curve is defined by plotting the average proportion of infected nodes (y-axis) against the seed node proportion (x-axis). Higher curves indicate greater vulnerability. In some cases, the area under this curve (AUC) was used as a synthetic vulnerability index. The function also optionally returns a ggplot-based visualization with confidence bands. Then, network vulnerability was defined as the relationship between the average percentage of reachable nodes and the percentage of seed nodes. This approach, based on the reachability from randomly selected nodes using ego-networks, was adapted from methods described by Natale et al. [9], Keeling and Eames [29], and Cardenas et al. [10], where vulnerability is expressed in terms of the system's structural capacity to propagate infection through network connections. The vulnerability was calculated within each region using monthly, static, directed networks. Movements to slaughterhouses, which usually have minimal impact on most directly transmitted diseases, were excluded from the analysis. To estimate network vulnerability, a fixed proportion of nodes was randomly selected as seed nodes (i.e., premises assumed to be infected). For each proportion tested (0.02%, 0.10%, 0.18%, 0.26%, 0.34%, and 0.50% of all nodes), 1,000 independent simulations were performed using the defined vulnerability metric, to estimate the average vulnerability, along with its standard deviation (SD). The average vulnerability was mapped per month and region of each year, and the SD was used to calculate the 95% confidence interval of the vulnerability curve.

The vulnerability function was applied to the aggregated data. Then, the vulnerability of each region was plotted for each month of the years, from 2013 to 2022. The results of vulnerability at the lower seed percentage (0.02) by region and by month were used for the time series decomposition, employed to assess trend and seasonality components. Subsequently, a dissimilarity matrix was constructed based on the correlation between the vulnerability of Minas Gerais regions. Hierarchical cluster analysis was then conducted to identify regions with similar long-term vulnerability patterns. The correlation and cluster analysis were performed with scaled (in average and standard deviation) vulnerability values.

The results of vulnerability per month and per region were also tested for Pearson´s correlation significance against the following global network metrics: the giant strongly connected component ratio (GSCCR), the number of cliques, the size of the big cliques, the number of big cliques, the transitivity, the edges density, the average path length and the unreachability ratio (UR). Table 1 provides definitions and explanations of the above metrics.

## Global vulnerability (Super spatial spreaders)

The vulnerability of each region, as described in the previous section, was assessed by considering only the internal movements within each region. This vulnerability indicator provides an estimation of how many nodes could be affected by an outbreak once one or more nodes in the region become exposed or infected. To provide a comprehensive picture of the state's vulnerability, was developed a procedure that considers long-distance connections.

First, it was evaluated the distribution of edge distances at the state level for the year 2022, the most recent year analyzed. The year 2022 was chosen for the super spatial hubs analysis, since the previously years showed the same pattern in vulnerability and similar behavior in the network analysis performed elsewhere [14] and was the more recent year of the dataset. The threshold distance was determined with the seventy-five percentile of its distribution. Then, a specific function (S2 Function) was developed to calculate the weighted average distance (where the weight is the number of animals moved) and the strength for both incoming and outgoing movements for each network node. A simple example of these calculations for each node is shown in Fig 2. Links below the threshold distance were excluded from this calculation. This approach ensures that all nodes have a distance and strength value,

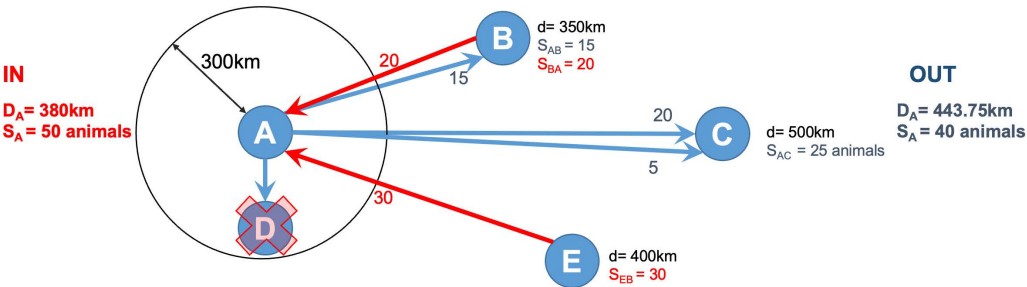

**Fig 2. Spatial hub metrics calculation for node A: the average distance (D$_A$) and strength (S$_A$) of incoming and outgoing animal movements.** D$_A$ is the weighted average distance based on the number of animals moved, while S$_A$ is the total number of animals moved. Red arrows indicate incoming movements, and blue arrows indicate outgoing movements, with labels showing distances and strengths. The movement to node D is excluded due to its distance from A being less than the 300 km threshold.

reflecting their behavior in terms of both incoming and outgoing long-distance trade. The Mahalanobis distance was used to identify nodes with high values in the joint distribution of distance and strength (greater than 75% on the log-transformed values) [30]. Nodes identified through this method are involved in the long-distance trade of many animals and are thus defined as spatial spreaders (or hubs). Spatial spreaders are further categorized into Spatial Spreaders (hub OUT), which are the origin of long-distance movements and can spread disease over a long-range, and Spatial Susceptible (hub IN), which are the destinations of movements originating from distant places and can introduce disease into a given region.

### Risk-based surveillance and interventions

The network vulnerability and spatial spreaders analyses were used to identify the region with higher vulnerability in terms of connections and with the greater number of super susceptible nodes in Minas Gerais state in 2022. To evaluate different intervention scenarios for network vulnerability reduction, several node metrics were considered. Specifically, the impact of eliminating 5% of the nodes with higher IN and OUT degree and higher betweenness centrality values, along with randomly selected nodes was assessed. In-degree, out-degree and betweenness centrality were selected for the intervention scenarios because they are widely used in veterinary epidemiology and have shown effectiveness in identifying high-risk nodes in livestock movement networks. These measures are especially relevant in directed movement networks, where flow direction matters [2,4,8,10].

The identification of super spatial susceptible and spreader nodes in the evaluated region enabled the detection of potential areas at risk for long-range introduction and diffusion. To quantify the sustainability of targeted surveillance based on previous years' rankings, the proportion of nodes with the highest value for the measure most significantly impacting regional vulnerability was evaluated over a nine-year period (years one to nine) (S3 Fig) Furthermore, the vulnerability decrease was evaluated by removing the nodes with higher betweenness centrality values in the network of the previous year (2021) from the network of 2022.

### Software

All analysis were conducted into R software version 4.3.0 [19], being the data organized with the packages "readxl" version 1.4.2 [20], "forecast" [21], "stringi" [22] and "tidyverse" [23]. All vulnerability analyses were performed with the packages "tidyverse" [23], "igraph" package [31] and "geobr" package [32]. Additionally, all spatial spreader analyses were performed with the packages "tidyverse" [23], "igraph" package [31], "geobr" package [32] and "geosphere" [33].

## Results

### Local vulnerability analysis inside regions

The vulnerability analysis per region and per month of the network from cattle movement in Minas Gerais, Brazil, from 2013 to 2022, showed the Triângulo Mineiro/ Alto Paranaíba (05) as the most vulnerable except in July of 2017 and 2022, when Campo das Vertentes (11) region showed higher vulnerability, based on the threshold greater than 0.02% of seed used for the analysis. Over the years, 2020 stood out as the year with the lowest vulnerability across all regions in every month, compared with the years before and after. The regions that appeared less vulnerable throughout the months in all years were Jequitinhonha (03) and Metropolitan of Belo Horizonte (07). All regions appeared to increase in vulnerability during July throughout all analyzed years. The vulnerability by region and month for 2022 can be seen on Fig 3, while data for the years 2013–2021 can be found in S4 Fig.

The graph and map showing the mean vulnerability of the regions revealed a division of the state in two extremes of vulnerability, with the east and west regions of Minas Gerais showing distinct patterns (Fig 4). The cluster dendrogram based on the correlation of regions vulnerability identified two distinct clusters: one included Triângulo Mineiro/ Alto Paranaíba (05), South/ Southeast of Minas (10), Campo das Vertentes (11), Northwest (01) and West of Minas (09)

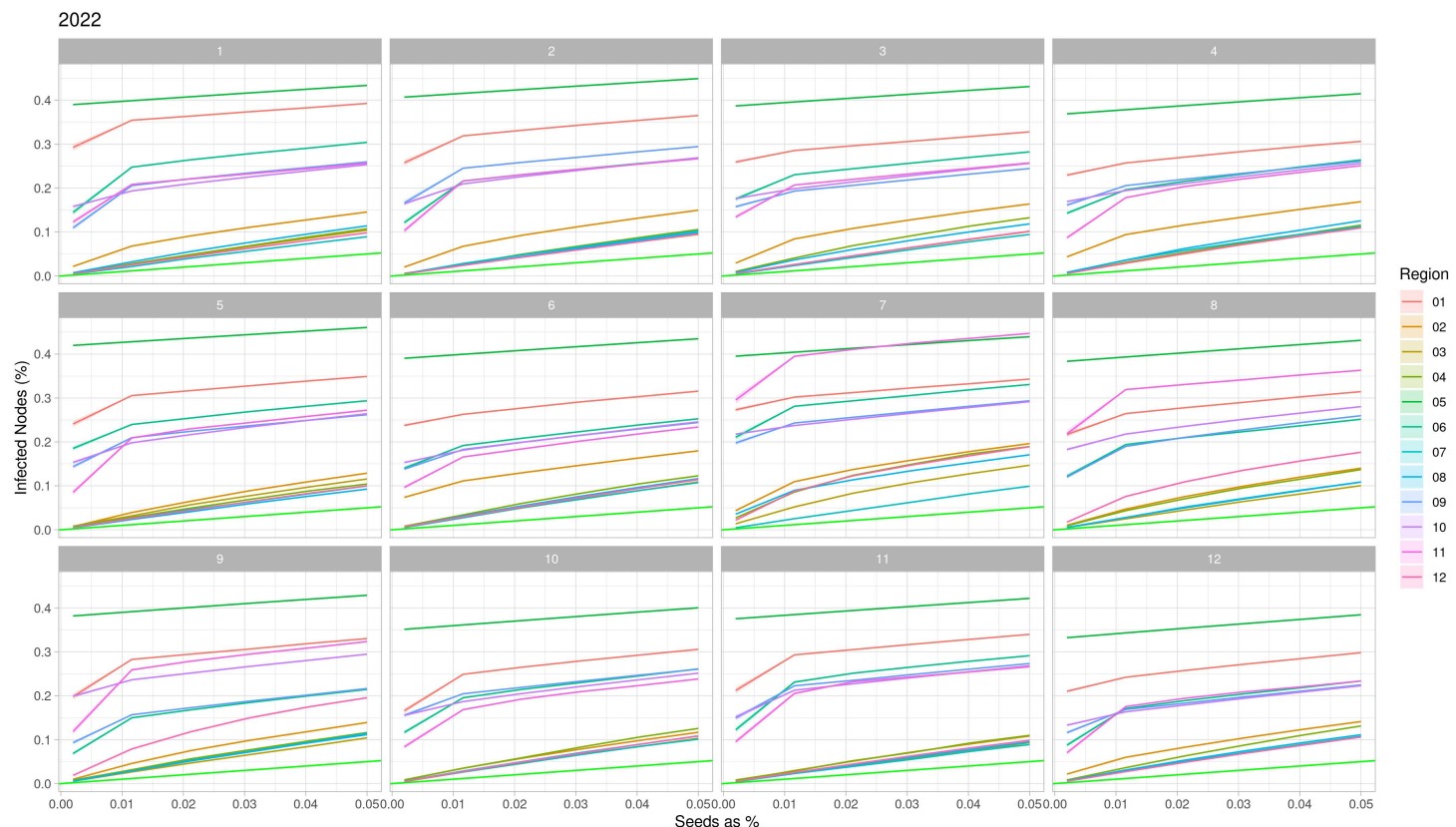

**Fig 3. Network Vulnerability of cattle movement per month and region in Minas Gerais state, Brazil in 2022.** The colored lines are the regions, and the light green represents the less possible vulnerability. The regions are 01: Northwest Minas, 02: North Minas, 03: Jequitinhonha, 04: Vale do Mucuri, 05: Triângulo Mineiro/Alto Paranaíba, 06: Central Minas, 07: metropolitan area of Belo Horizonte, 08: Vale do Rio Doce, 09: West Minas, 10: South/Southeast Minas, 11: Campo das Vertentes and 12: Zona da Mata.

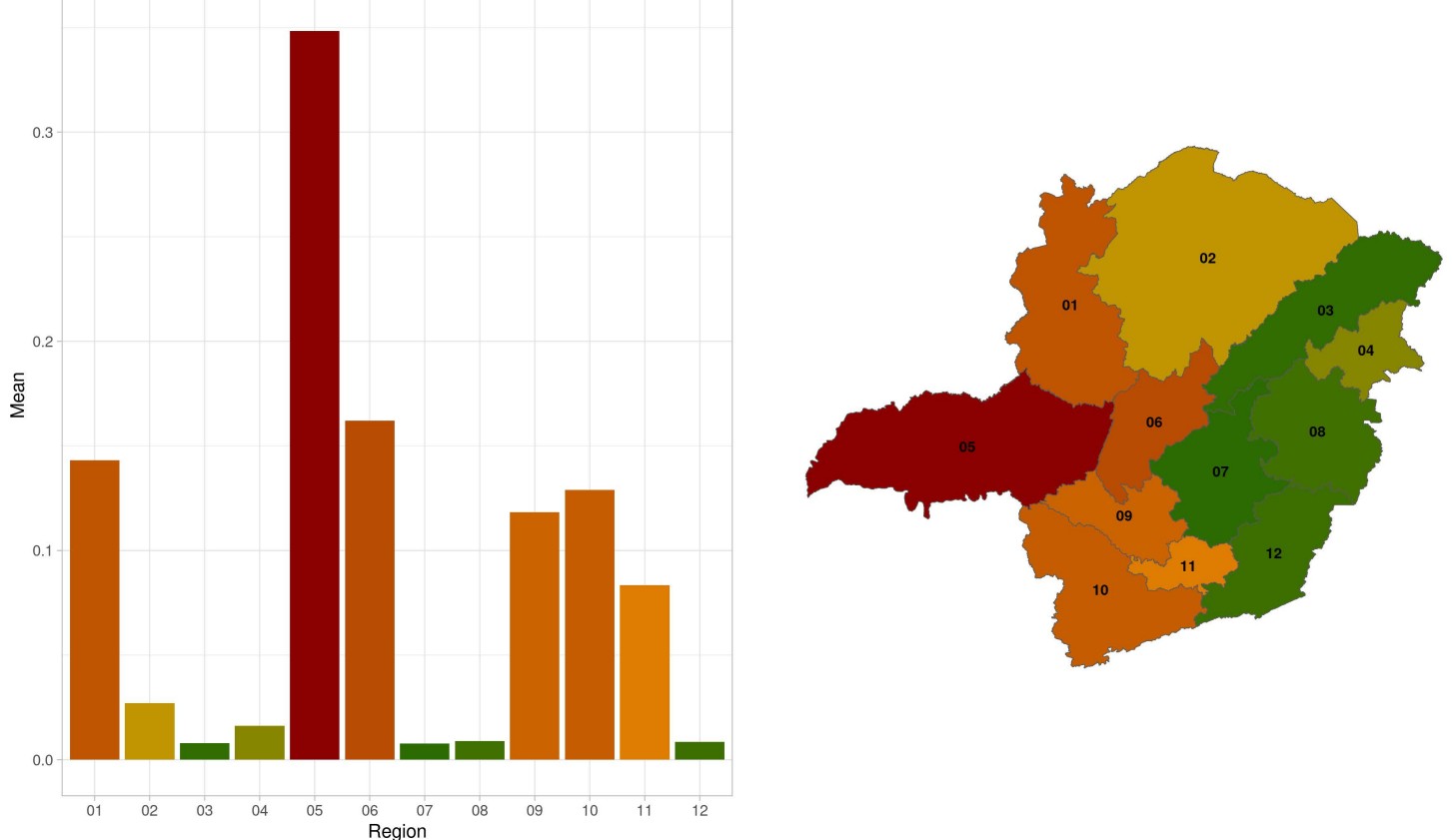

**Fig 4. Network vulnerability mean per region in Minas Gerais, Brazil, from 2013 to 2022.** The colors indicate greater values of vulnerability, dark red is more vulnerable and darker green less vulnerable. The regions are 01: Northwest Minas, 02: North Minas, 03: Jequitinhonha, 04: Vale do Mucuri, 05: Triângulo Mineiro/Alto Paranaíba, 06: Central Minas, 07: metropolitan area of Belo Horizonte, 08: Vale do Rio Doce, 09: West Minas, 10: South/ Southeast Minas, 11: Campo das Vertentes and 12: Zona da Mata.

regions, and the other cluster comprised the regions North (02), Jequitinhonha (03), Vale do Mucuri (04), Central of Minas (06), Metropolitan of Belo Horizonte (07), Vale do Rio Doce (08) and Zona da Mata (12) (Fig 5).

The vulnerability trend per region demonstrated very different intensity changes, however all regions decreased in 2020. The regions Jequitinhonha (03), Triângulo Mineiro/Alto Paranaíba (05), Central (06), South/Southeast Minas (10) and Campo das Vertentes (11) showed a strong increment in the vulnerability after 2021, in contrast to the other regions where the trend was very stable (Fig 6). On the other hand, the vulnerability seasonality per region showed different patterns among the regions, with peaks around June/ July, except for Triângulo Mineiro/ Alto Paranaíba (05), where no considerable peaks occurred (Fig 7). Noteworthy, in November most regions experienced a negative peak, except for the Northwest (01), North (02), Central (06) and Zona da Mata (12) regions.

All tested measures (GSCCR, the number of cliques, the size of the big cliques, the number of big cliques, the transitivity, the edges density, the average path length and the unreachability ratio (UR)) were significantly correlated with vulnerability. The GSCC size and the unreachability ratio (UR) exhibited the strongest positive and negative correlation values (0.94 and −0.97, respectively) (Table 2). The correlation matrix between the descriptive measures can be found in the S5 Fig. A summary of the descriptive measures of the network per region can be found in Table 3.

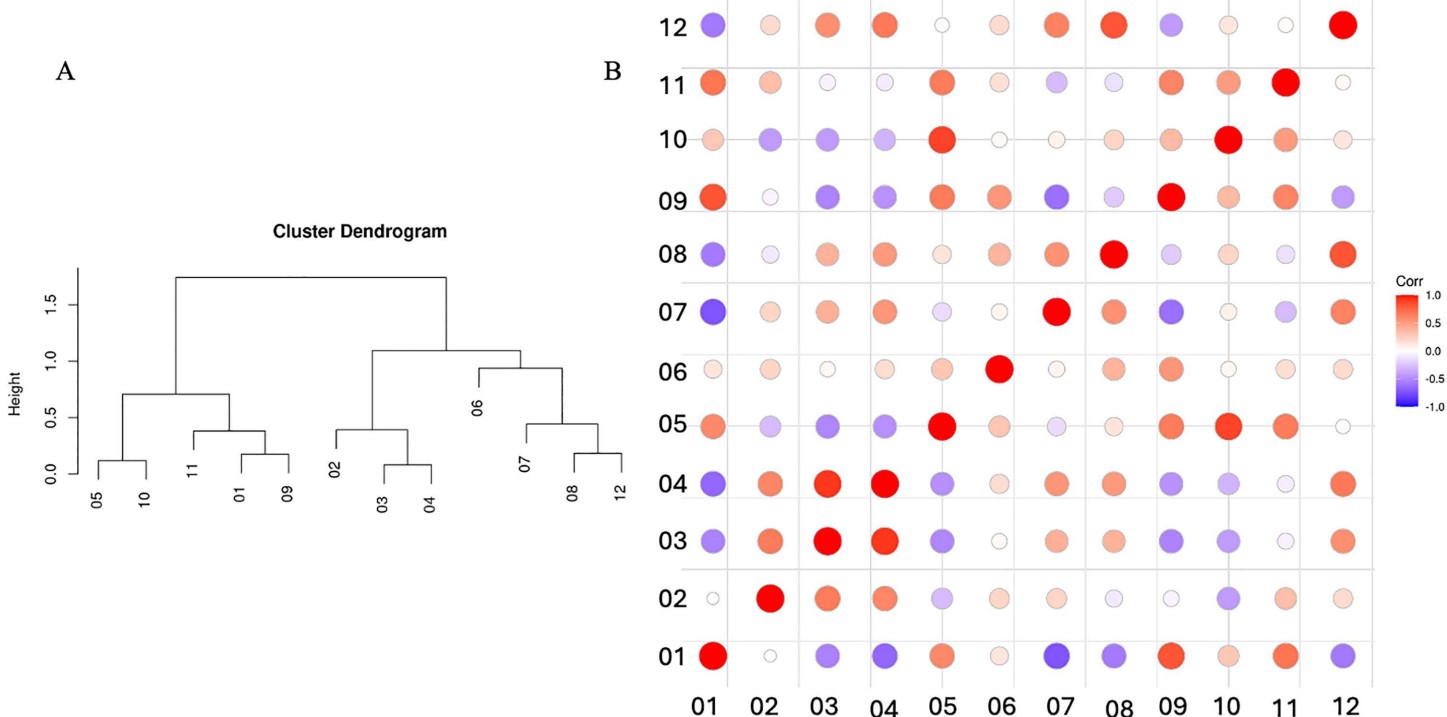

**Fig 5. Vulnerability correlation between regions of Minas Gerais, Brazil. A)** Cluster dendrogram build with the correlation between the vulnerability of each region. **B)** Colored matrix of regional vulnerability correlation, showing positive correlations (red) and negative correlations (magenta), the size of the circles indicates the magnitude of the correlation. The regions are 01: Northwest Minas, 02: North Minas, 03: Jequitinhonha, 04: Vale do Mucuri, 05: Triângulo Mineiro/Alto Paranaíba, 06: Central Minas, 07: metropolitan area of Belo Horizonte, 08: Vale do Rio Doce, 09: West Minas, 10: South/Southeast Minas, 11: Campo das Vertentes and 12: Zona da Mata.

## Global vulnerability (Super Spatial Hub)

The threshold found using the distances distribution resulted in 300 km. The number of nodes trading locally, with all movements occurring within the threshold, was 189,522 and 237,548, for incoming and outgoing movements, respectively. The number of nodes receiving/sending no animals at all were 127,550 and 72,968, respectively. Nodes moving cattle over the threshold were 8,844 (receiving) and 15,400 (sending).

The analysis identified 476 spatial spreaders and 332 spatial susceptible nodes. Overall, the hubs were concentrated between 2 and 3 log strength (corresponding to 100 and 1000 animals) and the log distance ranged from 5.72 to above 6.5 (corresponding to 525 and 665.14 km), for both directions of movement (IN and OUT) (Fig 8). Considering the hub´s geographic distribution per region, those receiving animals from farther nodes were in Triângulo Mineiro/ Alto Paranaíba (05), while the region sending animals to faraway distances was the region Vale do Mucuri (04) (Fig 8).

The hubs ratio showed the Vale do Mucuri (04) as the region with greater values in both directions (IN, OUT), and the North region was the one with the major value of strength ratio, transporting more animals in relation to its population compared to the other regions, in both directions IN and OUT (S6 Fig). The number of nodes removed because of the wrong geolocation were 8,181 (1.33) from the remaining nodes without slaughterhouse movements (614,372).

## Risk-based surveillance and interventions

The vulnerability analysis revealed region 05 (Triângulo Mineiro/ Alto Paranaíba) as the most vulnerable among all regions in Minas Gerais. The same region showed also the presence of many super susceptible nodes, making it more vulnerable

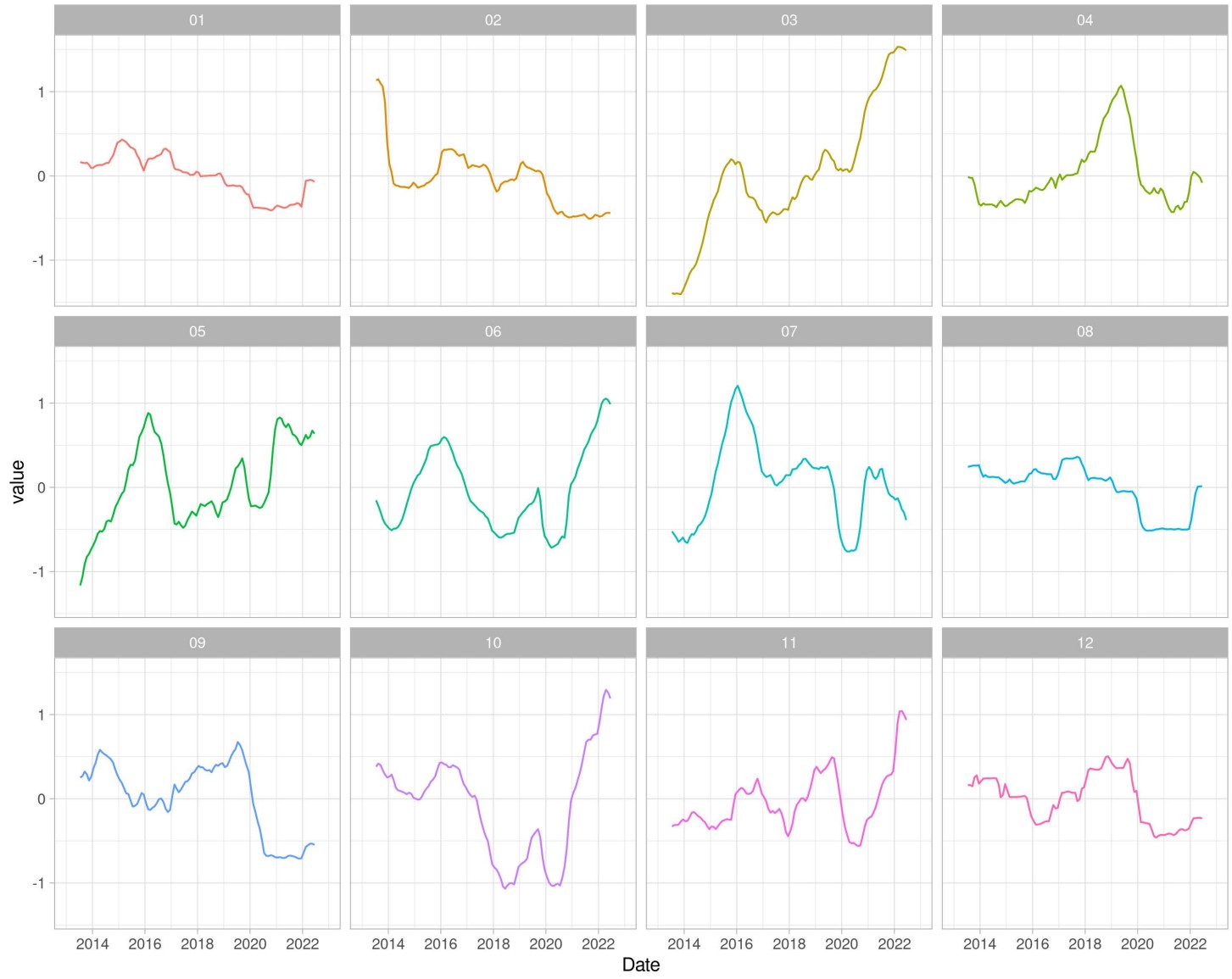

**Fig 6. Vulnerability trend per region per year, from 2013 to 2022.** The regions are 01: Northwest Minas, 02: North Minas, 03: Jequitinhonha, 04: Vale do Mucuri, 05: Triângulo Mineiro/Alto Paranaíba, 06: Central Minas, 07: metropolitan area of Belo Horizonte, 08: Vale do Rio Doce, 09: West Minas, 10: South/Southeast Minas, 11: Campo das Vertentes and 12: Zona da Mata.

to disease introduction from all over the state. Given these characteristics, we focused on this region to evaluate and find the most effective targeted surveillance or intervention scenario. The risk-based surveillance and intervention analysis revealed that, among the four tested scenarios, interventions targeting nodes with higher betweenness centrality significantly reduced network vulnerability compared to the original graph. Interventions targeting nodes with higher out-degree values also led to a notable decrease in vulnerability. These interventions proved to be more effective than those excluding nodes with higher in-degree or randomly chosen. More details can be found in Fig 9.

The number of super susceptible found at Triângulo Mineiro/ Alto Paranaíba (05) was 116 premises and number of super spreaders was 47 in the region, the number of nodes with both classifications was 13 nodes. Fig 10 highlights the

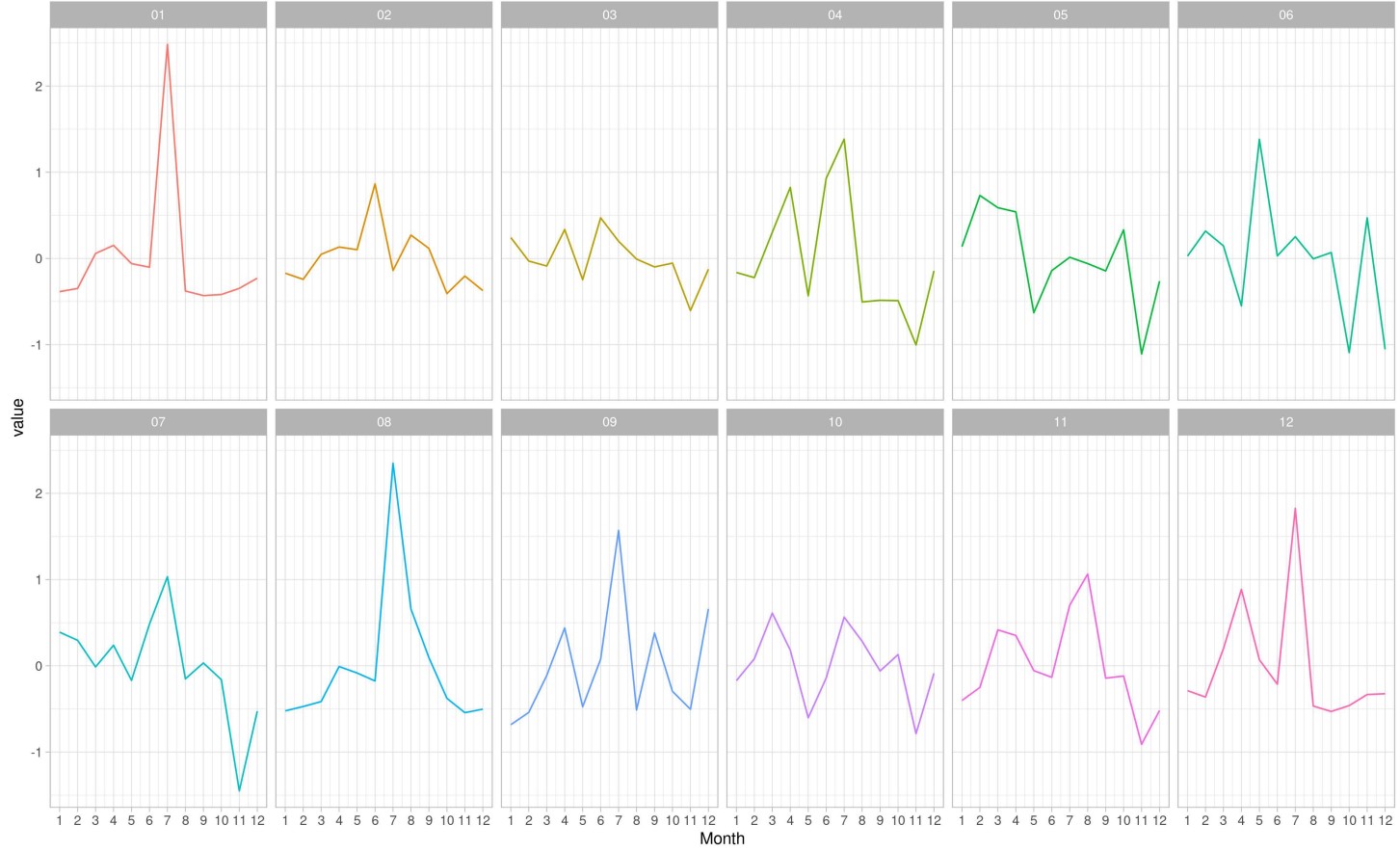

**Fig 7. Vulnerability seasonality per region and per month of 2022.** The regions are 01: Northwest Minas, 02: North Minas, 03: Jequitinhonha, 04: Vale do Mucuri, 05: Triângulo Mineiro/Alto Paranaíba, 06: Central Minas, 07: metropolitan area of Belo Horizonte, 08: Vale do Rio Doce, 09: West Minas, 10: South/Southeast Minas, 11: Campo das Vertentes and 12: Zona da Mata.

**Table 2. Pearson Correlation between network vulnerability per region and per month of cattle movement and descriptions measures of the cattle movement network in Minas Gerais, Brazil, from 2013 to 2022.**

| Connectivity measure | Pearson Correlation | p-value | 95% CI |
|---|---|---|---|
| Giant strongly component size ratio | 0.94 | <0.001 | 0.93 to 0.94 |
| Number of cliques | 0.73 | <0.001 | 0.70 to 0.75 |
| Size of the biggest clique | 0.56 | <0.001 | 0.53 to 0.60 |
| Number of bigger cliques | 0.08 | 0.002 | 0.03 to 0.13 |
| Transitivity | −0.47 | <0.001 | −0.51 to −0.43 |
| Edge density | −0.22 | <0.001 | −0.26 to −0.16 |
| Average path length | 0.68 | <0.001 | 0.65 to 0.71 |
| Unreachability ratio | −0.97 | <0.001 | −0.98 to −0.97 |

95%CI: Confidence interval at 95%.

**Table 3. Summary of the yearly network measures per regions of Minas Gerais, Brazil, from 2013 to 2022 (average values).**

| Region | Giant strongly component size ratio | Number of cliques | Size of the biggest clique | Number of bigger cliques | Transitivity | Edge density | Average path length | Unreachability ratio |
|---|---|---|---|---|---|---|---|---|
| Northwest Minas (01) | 0.08 | 3,914.13 | 3.47 | 35.49 | 0.01 | 0.00 | 5.43 | 0.95 |
| North Minas (02) | 0.01 | 7,394.53 | 3.33 | 37.11 | 0.02 | 0.00 | 4.55 | 1.00 |
| Jequitinhonha (03) | 0.01 | 2,165.89 | 3.09 | 11.34 | 0.02 | 0.00 | 1.99 | 1.00 |
| Vale do Mucuri (04) | 0.02 | 2,035.22 | 3.19 | 17.70 | 0.03 | 0.00 | 3.02 | 0.99 |
| Triângulo Mineiro/Alto Paranaíba (05) | 0.15 | 21,200.38 | 4.30 | 20.56 | 0.01 | 0.00 | 5.99 | 0.86 |
| Central Minas (06) | 0.10 | 3,366.16 | 3.46 | 35.46 | 0.01 | 0.00 | 5.01 | 0.93 |
| Metropolitan area of Belo Horizonte (07) | 0.01 | 2,056.30 | 3.08 | 14.31 | 0.03 | 0.00 | 2.21 | 1.00 |
| Vale do Rio Doce (08) | 0.01 | 4,745.69 | 3.18 | 24.81 | 0.02 | 0.00 | 2.51 | 1.00 |
| West of Minas (09) | 0.07 | 4,537.03 | 3.55 | 34.98 | 0.02 | 0.00 | 5.65 | 0.96 |
| South/Southeast of Minas (10) | 0.07 | 10,087.08 | 4.02 | 21.75 | 0.02 | 0.00 | 6.61 | 0.97 |
| Campo das Vertentes (11) | 0.09 | 1,774.20 | 3.38 | 14.55 | 0.02 | 0.00 | 4.58 | 0.96 |
| Zona da Mata (12) | 0.01 | 4,098.33 | 3.17 | 19.51 | 0.03 | 0.00 | 2.43 | 1.00 |

areas of the state connected to the region's super spatial hub. Particularly, it shows the zones of potential spread and introduction of a possible disease over long distances. From the super susceptible plus super spreaders nodes, 48.47% (79/163) were nodes excluded in the risk-based analysis considering the higher betweenness; 44.17% (72/163) of the spatial spreaders were nodes excluded due to high in degree, 12.88% (21/163) were nodes with greater out degree and 5.52% (9/163) of the super susceptible plus super spreaders nodes were among the nodes excluded at random in the risk-based analysis. Since node selection based on betweenness centrality includes a greater proportion of spatial hubs in the region compared to out-degree, betweenness centrality is the most suitable measure for reducing the region's vulnerability. As the ranking persistence analysis (S3 Fig) showed values around 50% with a one-year interval, the vulnerability assessment was repeated using the betweenness values from the previous year, and the comparison is shown in Fig 9 B.

## Discussion

This study highlights the significance of network and spatial dynamics analysis for enhancing animal disease control and prevention within the Minas Gerais state livestock sector. By identifying the most vulnerable regions (Triângulo Mineiro/Alto Paranaíba and Vale do Mucuri) the research indicated places for targeted interventions, which can substantially mitigate the risk of animal disease spread/introduction. Therefore, our findings support the development of more effective control/prevention strategies for managing animal infectious diseases in Minas Gerais, Brazil.

The adopted vulnerability measure provides an estimate of an epidemic spread due to the network's structure [29], by considering the number of nodes that can be reached from randomly chosen seed within the network. [29]. It is a precautionary measure, as it offers a pessimistic estimate of the potential epidemic size, which is influenced by other factors, such as the actual sequence of movements and epidemiological parameters of the disease. However, it allows the comparison of the vulnerability of different networks (being a ratio) without depending on a specific disease. The correlations shown in Table 2 confirm the coherence and validity of the considered measure, being strongly correlated with the GSCC and inversely correlated with the Unreachability Ratio. The internal vulnerability analysis revealed the most fragile regions exposed to the risk of disease introduction due to contact with other positive regions capable of infecting a high number of

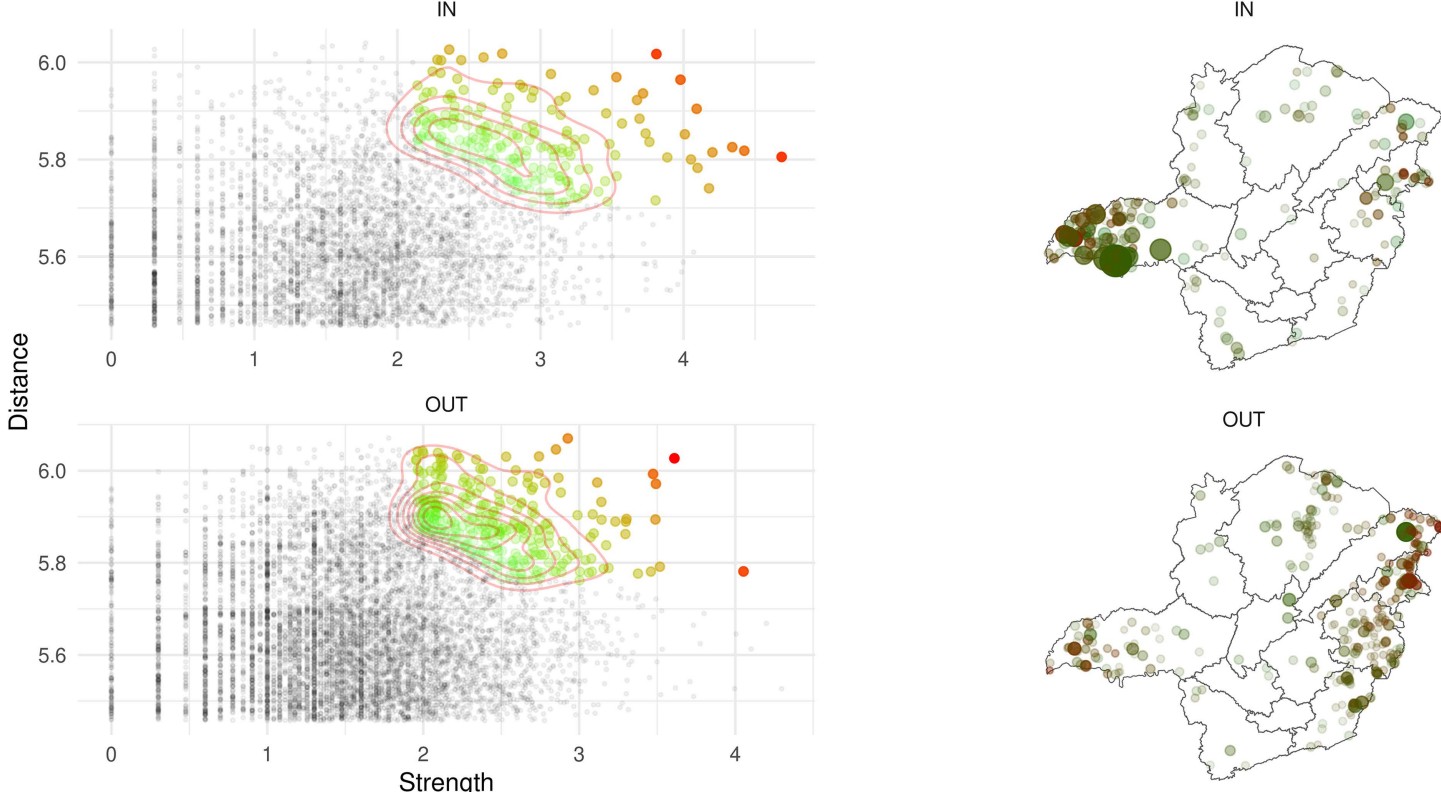

**Fig 8. Spatial spreaders (hubs) distribution.** In the left, ingoing and outgoing nodes distribution in relation to strength and distance. The green to red points represents the nodes that had Mahalanobis distance above the threshold, being the hubs, being green the lower distance and red the higher distance. On the right is the ingoing and outgoing geographic distribution of the hubs. The dark red color points are nodes that received or sent animals faraway. The transparency of the points was plotted in accordance with the Mahalanobis distance, and the size was considering the strength (number of transported animals).

properties, in the absence of control measures. Overall, the results have shown that the Triângulo Mineiro/ Alto Paranaíba region has been the most vulnerable over the years, with other regions varying in their vulnerability across the analyzed years (2013–2022), due to their sizes, connectivity and livestock production profile [34]. Due to being a spatial suscepti-ble, Triângulo Mineiro/ Alto Paranaíba is likely to become exposed to infection from faraway positive premises, and with its internal vulnerability could then rapidly spread the infection to most of the region's cattle herds. The greater vulnerability of Triângulo Mineiro/ Alto Paranaíba among the regions of Minas Gerais may be justified by high connectivity and the higher ratio of the GSCC among all the regions (Table 3). Indeed, the region exhibited more movement from 2013 to 2022, with a higher concentration of cattle population and livestock events [14,15] compared to the other regions. On the other hand, the regions with lower vulnerability (Jequitinhonha, Metropolitan of Belo Horizonte, Vale do Rio Doce and Zona da Mata) were regions with fewer cattle movements and smaller cattle populations [14,15], lower values of GSCC and fewer connections. The trend of vulnerability over time used for the correlation and cluster analysis suggests possible groups of regions with similar vulnerability behavior. Regions that are very close to each other in terms of vulnerability time series (Fig 5) were also found to be geographically closer (Fig 4). Indeed, whether a threshold dividing the dendrogram into two clusters is set, it splits the state into two contiguous macro areas: one with higher levels of vulnerability on the western side, and on the eastern side lower levels of vulnerability would be observed. Interestingly, the Central region, which lies geographically between these two clusters, would be included in the western group, despite having higher vulnerability.

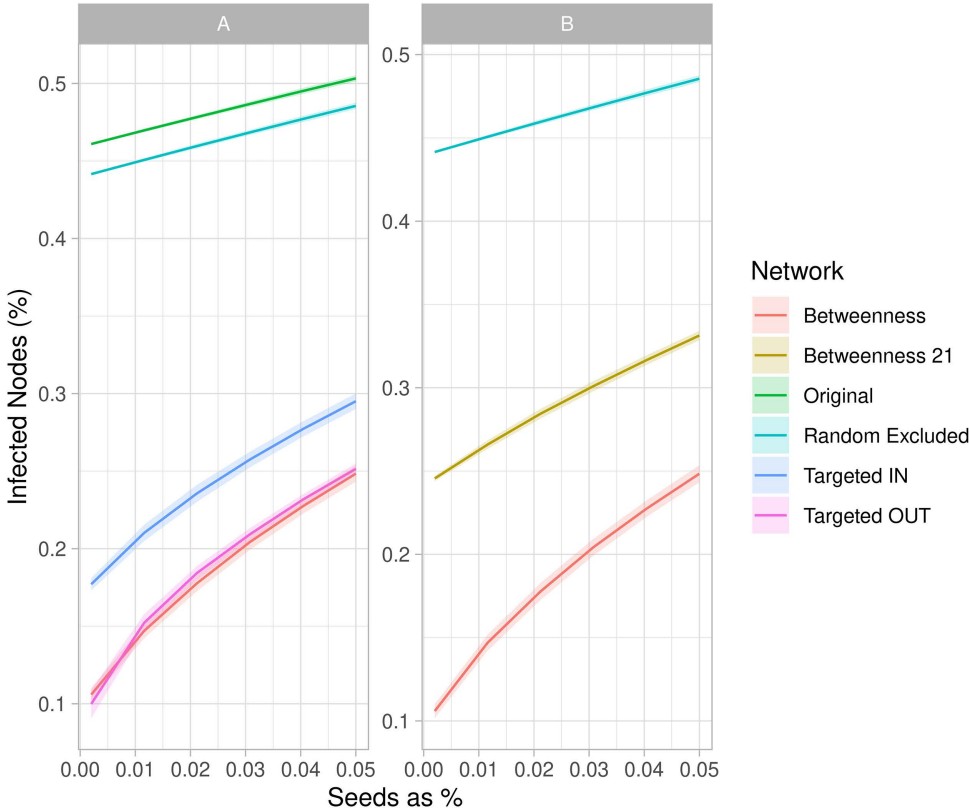

**Fig 9. Risk-Based Assessment of the Triângulo Mineiro/ Alto Paranaíba Region.** Reduction in vulnerability when 5% of nodes are eliminated based on higher betweenness centrality, higher in-degree, higher out-degree, and randomly for the year 2022 **(A)**. Impact on vulnerability when nodes are removed from the 2022 network based on the betweenness centrality values calculated from the 2021 network ('Betweenness 21'), compared to other strategies **(B)**.

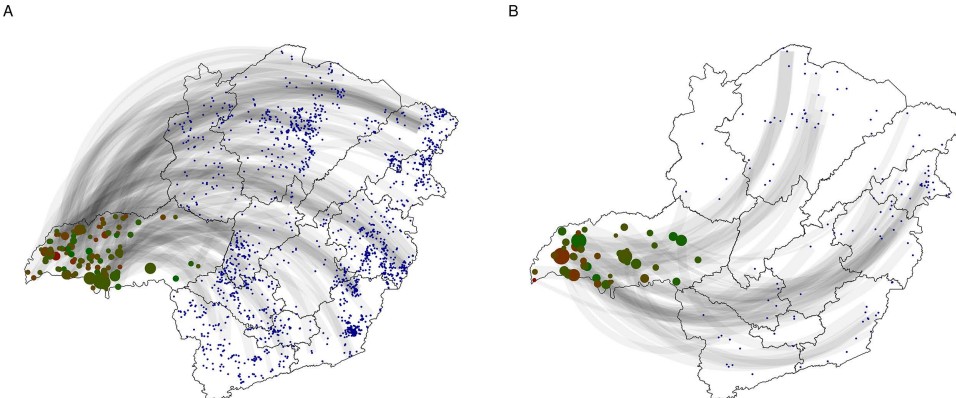

**Fig 10. A) Ingoing and B) outgoing hubs at Triângulo Mineiro/ Alto Paranaíba region in 2022.** The dark red points are the nodes receiving or sending animals far away and dark green are nodes receiving or sending animals to far livestock properties, however closer than the dark red points. The transparency of the points is the value of Malahanabis distance, and the size is the strength of the movements (number of transported animals). The blue points are the sending (outgoing movements) and the receiver (incoming movements) nodes. The grey lines are the movements, where the linewidth is the number of animals transported divided by 100. The darker the line the more the movements.

This suggests that it shares a similar temporal trend with the western group but generally exhibits higher values. This information is very important to plan interventions, such as blocking movements in some regions, transit and hubs surveillance, since it considers the vulnerability similarity of the regions composing each cluster.

In addition, the trend of network vulnerability (Fig 6) showed a decrease in vulnerability in 2020 compared to other years and across all regions, likely due to the SARS-Cov2 pandemic when the livestock events, such as fairs and auctions, were restricted [35], even though the movement among farms did not decrease [14]. Furthermore, the analysis of the vulnerability seasonality revealed a noticeable increase between May and July, the months when livestock events are concentrated in Minas Gerais state, and a decrease in the following months, with a little increase in November (Fig 7). The variation in seasonality coincided with the foot-and-mouth disease vaccination period in Minas Gerais, possibly due to the mandatory vaccination required for GTA issuance to move the animals in Brazil [36].

Moreover, the correlations exhibited in Table 2 with measures of network connectivity highlighted the options to be considered for targeted interventions in the whole network of Minas Gerais. For instance, our results showed that networks characterized by larger giant strongly connected components, a higher number of cliques, larger clique sizes, and longer average path lengths tend to exhibit higher vulnerability to disease spread. These findings highlight the importance of understanding network structure for effective disease control and prevention strategies.

The spatial spreaders analysis (Fig 8) identified several nodes within the state that consistently receive large numbers of animals from distant locations, termed "super spatial susceptible." The region identified as having the highest concentration of these nodes was Triângulo Mineiro/ Alto Paranaíba. These nodes are often associated with livestock events and cattle feedlots, experience multiple instances of animal arrivals from various livestock properties across the state, thereby increasing the risk of introducing diseases from distant, infected regions. Conversely, nodes characterized by extensive long-range outgoing connections, referred to as "super spatial spreaders," represent critical areas where, in the event of an infection, the disease could potentially disseminate across the entire state. The eastern part of the state, particularly the Vale do Mucuri region, was found to have a high density of super spatial spreader premises. These nodes are therefore of strategic importance for implementing tailored, risk-based interventions, towards the mitigation of risks by prioritizing these hubs for specific interventions according to their classification (spatial susceptible or spatial spreader). For instance, any disease that could be vaccinated against could concentrate efforts of vaccination of animals in these key areas, or yet, restrictions could be imposed on the importation of cattle from regions known to be positive. By strategically fragmenting the network, the introduction and spread of disease could be effectively controlled.

In fact, the risk-based analysis of network vulnerability focused on the Triângulo Mineiro/ Alto Paranaíba region, since it emerged as the most vulnerable in Minas Gerais, both in terms of internal susceptibility and external disease introduction. The analysis demonstrated that by applying centrality measures, such as betweenness, out-degree and in-degree in this order, to identify and exclude certain nodes, the region's vulnerability could be significantly reduced (Fig 9). This reduction in vulnerability was attributed to the fragmentation of the network, achieved by eliminating the more highly connected nodes [10]. In addition, the comparison between spatial spreaders and node elimination in the vulnerability analysis revealed a significant overlap of nodes identified by both methods. This reinforces the effectiveness of these analytical techniques and highlights the success of targeted interventions over random approaches.

Therefore, the present findings support the development of more effective control strategies for managing infectious diseases in Minas Gerais, Brazil, since vulnerability and spatial spreader analyses have been revealed as great tools for improving control and surveillance programs. They pinpoint the most important livestock properties for interventions, validating the advantages of targeted measures in reducing network vulnerability. Furthermore, it is important to highlight the significance of Triângulo Mineiro/ Alto Paranaíba and Vale do Mucuri regions in applying restriction of movements, as well as vaccination to contain the spread of infectious disease in Minas Gerais, Brazil.

One limitation of this study is the exclusion of data points with incorrect geolocation values, which were removed prior to the analysis. However, these excluded values constituted less than 10% of the entire dataset, allowing the integrity of

the study to be maintained. Another limitation was the use of data from 2022 instead of 2023, due to data availability constraints. Nevertheless, the ranking persistence analysis demonstrated a consistent pattern across years, suggesting that the findings are likely applicable to future years with similar results.

## Conclusion

In conclusion, our results showed the regions with more variability and the premises that were super spreaders and super susceptible, revealing targeted risk-based surveillance and intervention strategies to be used to improve disease control programs in Minas Gerais state, Brazil. Additionally, it is important to highlight the significance of Triângulo Mineiro/ Alto Paranaíba and Vale do Mucuri regions for cattle movement among the state and their importance when applying measures for containing infectious disease spread.

## Supporting information

**S1 Function. Network Vulnerability function of cattle movement per month, region and year in Minas Gerais, Brazil, from 2013 to 2022.** https://github.com/anninhactrbc/ntw_vulnerability_super_spreaders_susceptible.git.
(PDF)

**S2 Function. Super spreaders and super susceptible function.** https://github.com/anninhactrbc/ntw_vulnerability_super_spreaders_susceptible.git.
(PDF)

**S3 Fig. The proportion of nodes with the highest value of betweenness that most significantly contributes to reducing the region's vulnerability evaluated over the following years (from one to nine) to quantify the sustainability of targeted surveillance based on the ranking of previous years.**
(PDF)

**S4 Fig. Network Vulnerability of cattle movement per month and region in Minas Gerais state, Brazil in 2022.** The colored lines are the regions, and the light green represents the less possible vulnerability. The regions are 01: Northwest Minas, 02: North Minas, 03: Jequitinhonha, 04: Vale do Mucuri, 05: Triângulo Mineiro/Alto Paranaíba, 06: Central Minas, 07: metropolitan area of Belo Horizonte, 08: Vale do Rio Doce, 09: West Minas, 10: South/Southeast Minas, 11: Campo das Vertentes and 12: Zona da Mata.
(PDF)

**S5 Fig. Correlation Matrix of the network description measures of the network of cattle movement in Minas Gerais, Brazil from 2013 to 2022.**
(PDF)

**S6 Fig. Hubs IN and OUT distribution over hubs ratio and strength ratio.** A) Hubs IN distribution in the colored points. Grey scale color of the regions is the hub ratio in each region. B) Hubs OUT distribution in the colored points. Grey scale color of the regions is the hub ratio in each region. C) Hubs IN distribution in the colored points. Grey scale color of the regions is the strength ratio in each region. D) Hubs OUT distribution in the colored points. Grey scale color of the regions is the strength ratio in each region.
(PDF)

## Author contributions

**Conceptualization:** Anna Cecília Trolesi Reis Borges Costa, Lara Savini, Luca Candeloro.

**Data curation:** Anna Cecília Trolesi Reis Borges Costa.

**Formal analysis:** Anna Cecília Trolesi Reis Borges Costa, Lara Savini, Luca Candeloro.

**Funding acquisition:** Andrey Pereira Lage, Elaine Maria Seles Dorneles.

**Investigation:** Anna Cecília Trolesi Reis Borges Costa, Luca Candeloro.

**Methodology:** Anna Cecília Trolesi Reis Borges Costa, Luca Candeloro.

**Resources:** Elaine Maria Seles Dorneles.

**Software:** Anna Cecília Trolesi Reis Borges Costa, Luca Candeloro.

**Supervision:** Lara Savini, Luca Candeloro.

**Validation:** Anna Cecília Trolesi Reis Borges Costa, Luca Candeloro.

**Visualization:** Anna Cecília Trolesi Reis Borges Costa, Luca Candeloro.

**Writing – original draft:** Anna Cecília Trolesi Reis Borges Costa.

**Writing – review & editing:** Lara Savini, Luciana Faria de Oliveira, Andrey Pereira Lage, Elaine Maria Seles Dorneles, Luca Candeloro.

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
