## [Decision Letter · Decision Letter 0]

26 Mar 2025

Dear Dr. Dorneles,

We look forward to receiving your revised manuscript.

Kind regards,

Lucas D. B. Faria

Academic Editor

PLOS ONE

Journal Requirements:

3. Thank you for stating the following financial disclosure: The authors would like to thank the  the Coordenação de Aperfeiçoamento de Pessoal de Nível Superior (CAPES), Fundação de Amparo à Pesquisa do Estado de Minas Gerais (Fapemig) (RED 000132-22) and Conselho Nacional de Desenvolvimento Científico e Tecnológico (CNPq) (402774/2022-1), Brazil, for the financial support. EMSD is thankful to CNPq for her fellowship. 

4. Thank you for stating the following in the Acknowledgments Section of your manuscript: The authors would like to thank the Coordenação de Aperfeiçoamento de Pessoal de Nível Superior, Brasil (Capes), Fundação de Amparo à Pesquisa do Estado de Minas Gerais (Fapemig) (RED 000132-22) and Conselho Nacional de Desenvolvimento Científico e Tecnológico (CNPq) (402774/2022-1), Brazil, for the financial support. EMSD is thankful to CNPq for her fellowship.

Please remove any funding-related text from the manuscript and let us know how you would like to update your Funding Statement. Currently, your Funding Statement reads as follows: The authors would like to thank the  the Coordenação de Aperfeiçoamento de Pessoal de Nível Superior (CAPES), Fundação de Amparo à Pesquisa do Estado de Minas Gerais (Fapemig) (RED 000132-22) and Conselho Nacional de Desenvolvimento Científico e Tecnológico (CNPq) (402774/2022-1), Brazil, for the financial support. EMSD is thankful to CNPq for her fellowship.

5. In the online submission form, you indicated that all data will be made avaliable uppon request to the corresponding authors.

6. We note that Figure 1, 8 and 10 in your submission contain [map/satellite] images which may be copyrighted. All PLOS content is published under the Creative Commons Attribution License (CC BY 4.0), which means that the manuscript, images, and Supporting Information files will be freely available online, and any third party is permitted to access, download, copy, distribute, and use these materials in any way, even commercially, with proper attribution. For these reasons, we cannot publish previously copyrighted maps or satellite images created using proprietary data, such as Google software (Google Maps, Street View, and Earth). For more information, see our copyright guidelines: http://journals.plos.org/plosone/s/licenses-and-copyright.

a. You may seek permission from the original copyright holder of Figure 1, 8 and 10 to publish the content specifically under the CC BY 4.0 license.  

Additional Editor Comments:

Dear Author,

I apologize for the delay, but it has been challenging for us editors to find available collaborators for review.

Please consider the comments from the reviewer, who pointed out relevant issues to be addressed.

Sincerely,

Reviewers' comments:

Reviewer's Responses to Questions

**Comments to the Author**

1. Is the manuscript technically sound, and do the data support the conclusions?

Reviewer #1: Yes

2. Has the statistical analysis been performed appropriately and rigorously?

Reviewer #1: No

3. Have the authors made all data underlying the findings in their manuscript fully available?

Reviewer #1: No

4. Is the manuscript presented in an intelligible fashion and written in standard English?

Reviewer #1: Yes

Reviewer #1: General Comment

This study focuses on analyzing the vulnerability of livestock movement networks in Minas Gerais, Brazil. Such research is particularly valuable for anticipating the spread of animal diseases transmitted directly between animals. The objective is to identify the best strategies to reduce the network's disease vulnerability. According to the results, centrality measures such as out-degree and betweenness seem to be the best strategy. Additionally, the study highlights the most vulnerable regions within the state, which can be helpful for targeted surveillance efforts. However, the study does not fully capture the dynamics of an epidemic.

There is a lack of context regarding livestock movements, which would help justify the necessity of this type of study and guide the reader more effectively.

Certain sections lack well-organized ideas and sentences, making them difficult to follow. For instance, in the Materials and Methods section, some justifications for specific choices appear later in the text when they would be more helpful earlier.

The discussion section should also be improved to provide more context to justify the results (if possible). Additionally, it would be interesting to include perspectives on this study's potential implications and applications.

Specific comment

Abstract & Introduction

Comment 1 :

Add a sentence at the beginning of the abstract that briefly defines the topic.

Comment 2:

The introduction lacks sufficient context about the movements in Brazil, particularly in Minas Gerais. It is strongly recommended that a general overview of these movements and the diseases they transmit be provided, specifically focusing on the study area.

M&M

Comment 3 :

In the Materials & Methods section, adding a subsection titled "Software" and listing all the packages used would be better

Comment 4 :

It would be better to start the Vulnerability section (M&M) with a brief definition of network vulnerability.

Comment 5:

“This procedure was repeated 1000 times for each percentage of seed nodes (0.02%, 0.10%, 0.18%, 0.26%, 0.34%, and 0.50%) to estimate average and standard deviation (SD)”.

Do the percentages here correspond to the random percentage of selected nodes?

The following paragraph needs to be rewritten or clarified.“To estimate network vulnerability, a defined percentage of nodes was randomly selected as seed nodes and the igraph's ego function was used to calculate the number of nodes reachable downstream from each seed node, expressed as a percentage of the network size. This procedure was repeated 1000 times for each percentage of seed nodes (0.02%, 0.10%, 0.18%, 0.26%, 0.34%, and 0.50%) to estimate the average and standard deviation (SD)”

Comment 6:

Reference for this definition?

“Network vulnerability was then defined as the relationship between the average percentage of reachable nodes and the percentage of seed nodes. »

Comment 7:

This section should be moved to the part where you discuss the data. In this paragraph, you only need to specify that this was applied to the aggregated data. Additionally, in the data section, it would be helpful to include more details about what was used to reconstruct the networks.

“The datasets were grouped by local of origin, local of destination, region (origin and

destination) and month per year to create a list of the network objects, from which the

vulnerability R function (S1 function) was applied.”

Comment 8:

Repetition

“The vulnerability of each region was plotted for each month of the year, from 2013 to 2022”

“was plotted per month and region of each year”

Comment 9

Maybe you should mention it first, in the part where you talk about vulnerability.

“This vulnerability indicator provides an estimation of how many nodes could be affected by an outbreak once one or more nodes in the region become exposed or infected.”

Comment 10:

Suggestion: In the Vulnerability section, you can add two sub-sections: Local Vulnerability and Global Vulnerability.

Comment 11

Why was 2022 chosen for the distance evaluation?

The answer is provided later; I just saw it. It might be better to justify this point when you first mention it.

Comment 12

Justify why in-degree, out-degree and betweenness were selected as centrality measures in the tested scenarios (include references or other justifications). Why not eigenvector, for exemple?

Suggestions: It would be interesting to test multiple percentages to determine how many nodes need to be removed to achieve an optimal result. I suppose the higher the percentage, the more optimal the outcome. But what about the range between 1% and 5%? In other words, is removing 1% of nodes sufficient, or do we need to go up to 5%?

Comment 13 :

You need to read this subsection in its entirety to understand why the year 2022 was selected. Find a way to rewrite it so that the reader can grasp why the transition is made from the years 2013-2022 to 2022. Justify this at the beginning.

Results

Comment 14:

A figure is needed to illustrate this.

“The vulnerability analysis per region and per month of the network from cattle movement in Minas Gerais, Brazil, from 2013 to 2022, showed the Triângulo Mineiro / Alto Paranaíba (05) as the most vulnerable except in July of 2017 and 2022, when Campo das Vertentes (11) region showed higher vulnerability, based on the threshold greater than 0.02% of seed used for the analysis. Over the years, 2020 stood out as the year with the lowest vulnerability across all regions in every month, compared with the years before and after. The regions that appeared less vulnerable throughout the months in all years were Jequitinhonha (03) and Metropolitan of Belo Horizonte (07)”

Comment 15:

It is better to put this sentence in method “267 The year 2022 was chosen for the super spatial hubs analysis,since the previously years showed the same pattern in vulnerability and similar behavior in the network analysis performedelsewhere [19] and was the more recent year of the dataset”

Comment 16:

What does Betweenness 21 correspond to in Figure 9?

Discussion:

Rewrite this sentence: “The adopted vulnerability measure considers the number of nodes that can be reached from 349 randomly chosen nodes within the network. Assuming that the chosen nodes are disease seeds, 350 this measure provides an estimate of the epidemic due to the network's structure”

Comment 17:

I thought that out-degree and betweenness were the most important measures.

“The analysis demonstrated that by 420 applying centrality measures, such as in-degree, out-degree, and betweenness, to identify and 421 exclude certain nodes, the region's vulnerability could be significantly reduced”

**Do you want your identity to be public for this peer review?** For information about this choice, including consent withdrawal, please see our Privacy Policy

Reviewer #1: No

---

## [Author Response · Author response to Decision Letter 1]

9 Jun 2025

Dr. Lucas D. B. Faria

Academic Editor,

PLOS ONE

Thank you for your message regarding our manuscript “Network vulnerability of cattle movement in Minas Gerais, Brazil, from 2013 to 2022, PONE-D-24-59680. We would like to thank you for the appropriate and useful suggestions regarding our manuscript. All the points made by the reviewers are consistent and improved the manuscript.

Please find the enclosed Responses to Journal Requirements and reviewer #1.

Sincerely,

Prof. Elaine Maria Seles Dorneles

Departamento de Medicina Veterinária Preventiva,

Universidade Federal de Lavras

Av. Sul UFLA - Aquenta Sol - Lavras, Minas Gerais

Post office box: 3037

Postal Code: 37200-900. Brazil.

elaine.dorneles@ufla.br

Phone: + 55 35 3829 1712

Journal Requirements:

Answer: Thank you for the reminder. The files were all updated according to the style requirements.

Answer: Thank you for your request. The codes used in this paper are shared here https://github.com/anninhactrbc/ntw_vulnerability_super_spreaders_susceptible.git

3. Thank you for stating the following financial disclosure: The authors would like to thank the Coordenação de Aperfeiçoamento de Pessoal de Nível Superior (CAPES), Fundação de Amparo à Pesquisa do Estado de Minas Gerais (Fapemig) (RED 000132-22) and Conselho Nacional de Desenvolvimento Científico e Tecnológico (CNPq) (402774/2022-1), Brazil, for the financial support. EMSD is thankful to CNPq for her fellowship.

Answer: We appreciate your request. The correct finacial disclouser were corrected to:

“The authors would like to thank the Coordenação de Aperfeiçoamento de Pessoal de Nível Superior, Brasil (Capes), Fundação de Amparo à Pesquisa do Estado de Minas Gerais (Fapemig) (RED 000132-22) and Conselho Nacional de Desenvolvimento Científico e Tecnológico (CNPq) (402774/2022-1), Brazil, for the financial support. EMSD is thankful to CNPq for her fellowship. The funders had no role in study design, data collection and analysis, decision to publish, or preparation of the manuscript.”

4. Thank you for stating the following in the Acknowledgments Section of your manuscript: The authors would like to thank the Coordenação de Aperfeiçoamento de Pessoal de Nível Superior, Brasil (Capes), Fundação de Amparo à Pesquisa do Estado de Minas Gerais (Fapemig) (RED 000132-22) and Conselho Nacional de Desenvolvimento Científico e Tecnológico (CNPq) (402774/2022-1), Brazil, for the financial support. EMSD is thankful to CNPq for her fellowship.

Please remove any funding-related text from the manuscript and let us know how you would like to update your Funding Statement. Currently, your Funding Statement reads as follows: The authors would like to thank the Coordenação de Aperfeiçoamento de Pessoal de Nível Superior (CAPES), Fundação de Amparo à Pesquisa do Estado de Minas Gerais (Fapemig) (RED 000132-22) and Conselho Nacional de Desenvolvimento Científico e Tecnológico (CNPq) (402774/2022-1), Brazil, for the financial support. EMSD is thankful to CNPq for her fellowship.

Answer: Thank you for the correction. We removed the financial sources from the acknowledgement section.

5. In the online submission form, you indicated that all data will be made avaliable uppon request to the corresponding authors.

Answer: Thank you for your request. The data used for this project contain identification details of the premises such as name, latitude and longitude, used in the analysis, the availability of these information would have to be authorized by the owners and that is not possible to acquire in light of the great number of farms in Minas Gerais state, Brazil. Additionally, Brazil has a law Lei Geral de Proteção de Dados Pessoais (LGPD), Lei n° 13.709/2018, that prohibit us to share or even access of some the personal information that was in our data.

6. We note that Figure 1, 8 and 10 in your submission contain [map/satellite] images which may be copyrighted. All PLOS content is published under the Creative Commons Attribution License (CC BY 4.0), which means that the manuscript, images, and Supporting Information files will be freely available online, and any third party is permitted to access, download, copy, distribute, and use these materials in any way, even commercially, with proper attribution. For these reasons, we cannot publish previously copyrighted maps or satellite images created using proprietary data, such as Google software (Google Maps, Street View, and Earth). For more information, see our copyright guidelines: http://journals.plos.org/plosone/s/licenses-and-copyright.

a. You may seek permission from the original copyright holder of Figure 1, 8 and 10 to publish the content specifically under the CC BY 4.0 license.

Answer: We appreciate your concern, however the maps produced in this paper were developed by the authors. The based maps used the multipolygon that produce the limit lines of the state and country and it was aquired in a free R package called “geobr”, by Pereira R, Goncalves. _geobr: Download Official Spatial Data Sets of Brazil_. R package version 1.7.0. 2022, cited and referenced in the paper. This package uses free data on geographic information of Brazil and its states that are provided by Instituto Brasileiro de Geografia e Estatistica (IBGE). All information can be found here https://github.com/ipeaGIT/geobr.

Reviewer #1: General Comment

This study focuses on analyzing the vulnerability of livestock movement networks in Minas Gerais, Brazil. Such research is particularly valuable for anticipating the spread of animal diseases transmitted directly between animals. The objective is to identify the best strategies to reduce the network's disease vulnerability. According to the results, centrality measures such as out-degree and betweenness seem to be the best strategy. Additionally, the study highlights the most vulnerable regions within the state, which can be helpful for targeted surveillance efforts. However, the study does not fully capture the dynamics of an epidemic.

There is a lack of context regarding livestock movements, which would help justify the necessity of this type of study and guide the reader more effectively.

Certain sections lack well-organized ideas and sentences, making them difficult to follow. For instance, in the Materials and Methods section, some justifications for specific choices appear later in the text when they would be more helpful earlier.

The discussion section should also be improved to provide more context to justify the results (if possible). Additionally, it would be interesting to include perspectives on this study's potential implications and applications.

Specific comment

Abstract & Introduction

Comment 1:

Add a sentence at the beginning of the abstract that briefly defines the topic.

Answer: Thank you for your request. Please see the sentence at the beginning of the abstract.

“The analysis of network from cattle movements is an important approach to investigate areas and premises where diseases outbreaks can occur and be contained, being the vulnerability analysis a more profound comprehension of the network by combining the descriptive measures to the removal of nodes from the system into simulations that revel more vulnerable area and the best measure to be used while planning disease control.”

Comment 2:

The introduction lacks sufficient context about the movements in Brazil, particularly in Minas Gerais. It is strongly recommended that a general overview of these movements and the diseases they transmit be provided, specifically focusing on the study area.

Answer: We appreciate your suggestion. There is one paper already published describing the network of cattle movement in that region. Even so, we added a phrase in the introduction to give some context, as follow:

“The network analysis of cattle movement in Minas Gerais was shown elsewhere [14], describing a very connected network with movements more focused in the west and east sides of the state, being the Triângulo Mineiro / Alto do Paranaíba and Vale do Mucuri regions of great emphasis on movements and cattle population [15].”

M&M

Comment 3:

In the Materials & Methods section, adding a subsection titled "Software" and listing all the packages used would be better

Answer: We appreciate your suggestion. The new section was added to the Material and Methods.

“Software

All analysis were conducted into R software version 4.3.0[19], being the data organized with the packages “readxl” version 1.4.2 [20], “forecast” [21], "stringi" [22] and “tidyverse” [23]. All vulnerability analyses were performed with the packages “tidyverse” [23], “igraph” package [29] and “geobr” package [30]. Additionally, all spatial spreader analyses were performed with the packages “tidyverse” [23], “igraph” package [29], “geobr” package [30] and “geosphere” [32].”

Comment 4 :

It would be better to start the Vulnerability section (M&M) with a brief definition of network vulnerability.

Answer: We appreciate the suggestion. The definition of the vulnerability and the reason of its use were added to the beginning of the section about this subject.

“The vulnerability function considers the connections and the diameter of the network, considering more vulnerable the network that is more connected [10]. Additionally, vulnerability of a graph could be useful to show which measures are better to be considered for interventions inside the graph system, recognizing the leading nodes to be blocked in case of disease spread over the network [8–10], permitting the improvement of the use of human and financial resources in animal disease surveillance and control programs.”

Comment 5:

“This procedure was repeated 1000 times for each percentage of seed nodes (0.02%, 0.10%, 0.18%, 0.26%, 0.34%, and 0.50%) to estimate average and standard deviation (SD)”.

Do the percentages here correspond to the random percentage of selected nodes?

The following paragraph needs to be rewritten or clarified. “To estimate network vulnerability, a defined percentage of nodes was randomly selected as seed nodes and the igraph's ego function was used to calculate the number of nodes reachable downstream from each seed node, expressed as a percentage of the network size. This procedure was repeated 1000 times for each percentage of the seed nodes (0.02%, 0.10%, 0.18%, 0.26%, 0.34%, and 0.50%) to estimate the average and standard deviation (SD)”

Amswer: Thank you for your suggestion. To clarify the sentence, we added “randomly selected” just before “seed node” in the second part of the sentence. In the second sentence we added “random” before “seed nodes” just before the parenthesis, to clarify how the analysis were performed.

Comment 6:

Reference for this definition?

“Network vulnerability was then defined as the relationship between the average percentage of reachable nodes and the percentage of seed nodes. »

Answer: Thank you for your question. We understand your question and the confusion caused by the word “defined”, therefore, we removed the “defined as” to clarify that the vulnerability was identified based on the relationship explained in the above phrase.

Comment 7:

This section should be moved to the part where you discuss the data. In this paragraph, you only need to specify that t

---

## [Decision Letter · Decision Letter 1]

28 Aug 2025

Dear Dr. Dorneles,

Thank you for submitting your manuscript to PLOS ONE. After careful consideration, we feel that it has merit but does not fully meet PLOS ONE’s publication criteria as it currently stands. Therefore, we invite you to submit a revised version of the manuscript that addresses the points raised during the review process.

We look forward to receiving your revised manuscript.

Kind regards,

Lucas D. B. Faria

Academic Editor

PLOS ONE

Journal Requirements:

Additional Editor Comments :

Reviewers judged the manuscript high-quality and practically relevant to animal disease surveillance, praising its rigorous network-vulnerability approach (including spatial spreaders and intervention modeling), decade-long dataset, clear actionable identification of priority regions/metrics (Triângulo Mineiro/Alto Paranaíba; Vale do Mucuri; betweenness and out-degree), and open code.

Minor points to address in the revision among others pointed out by revision

Figure 9 (panel B) legend clarity

Please revise the legend to ensure it is fully self-explanatory. The following wording would be acceptable:

“Impact on vulnerability when nodes are removed from the 2022 network based on the betweenness centrality values calculated from the 2021 network (‘Betweenness 21’), compared to other strategies.”

Data Availability Statement (LGPD clarity)

The justification for restricted data access under the LGPD is appropriate. To maximize clarity, please state explicitly that data can be accessed for research purposes upon reasonable request and approval from the Instituto Mineiro de Agropecuária (IMA). For example:

“Data are available from the Instituto Mineiro de Agropecuária (IMA) (contact: ima@ima.mg.gov.br

) for researchers who meet the criteria for access to confidential data, as required by Brazilian law (Lei Geral de Proteção de Dados Pessoais — LGPD, Lei n° 13.709/2018).”

Please also ensure that the code repository link/DOI is cited in both the Data Availability Statement and the References (as appropriate).

Reviewers' comments:

Reviewer's Responses to Questions

**Comments to the Author**

Reviewer #2: All comments have been addressed

2. Is the manuscript technically sound, and do the data support the conclusions?

Reviewer #2: Yes

3. Has the statistical analysis been performed appropriately and rigorously?

Reviewer #2: Yes

4. Have the authors made all data underlying the findings in their manuscript fully available?

Reviewer #2: No

5. Is the manuscript presented in an intelligible fashion and written in standard English?

Reviewer #2: Yes

Reviewer #2: Reviewer Recommendation: Accept with Minor Revisions

Manuscript Number: PONE-D-24-59680R1

Title: Network vulnerability of cattle movement in Minas Gerais, Brazil, from 2013 to 2022

Corresponding Author: Elaine Maria Seles Dorneles, Ph.D.

Overall Recommendation:

This review strongly recommends the acceptance of the manuscript "Network vulnerability of cattle movement in Minas Gerais, Brazil, from 2013 to 2022" for publication in PLOS ONE, pending minor revisions. The authors have comprehensively and satisfactorily addressed the vast majority of the points raised by the reviewer in the previous round. The manuscript presents a robust and methodologically sound ecological and epidemiological analysis that makes a significant and applicable contribution to the field of animal disease surveillance and control.

The study's strengths are numerous:

Relevance and Applicability: The research addresses a critical need for evidence-based, risk-driven strategies to manage infectious diseases in livestock, a topic of global importance for food security, animal welfare, and economic stability.

Methodological Rigor: The application of network vulnerability analysis, combined with spatial spreader identification and risk-based intervention modeling, represents a sophisticated and powerful approach. The use of a decade-long dataset provides a robust basis for the analysis.

Clear Findings: The results clearly identify high-risk regions (Triângulo Mineiro / Alto Paranaíba and Vale do Mucuri) and the most effective network metrics (betweenness centrality and out-degree) for targeted intervention, providing actionable intelligence for veterinary services.

Transparency and Reproducibility: The authors have made their code publicly available, adhering to best practices for open science, which greatly enhances the value and reproducibility of the research.

The authors' response to the reviewers demonstrates a serious commitment to improving the manuscript. The revisions have enhanced the clarity of the methods, provided better context in the introduction, and improved the flow and justification of the analytical choices.

Addressing Previous Reviews:

The authors have successfully addressed the core concerns raised by Reviewer #1:

Context and Justification: The addition of context regarding cattle movements in Minas Gerais in the introduction is appropriate and sufficient.

Clarity and Organization: The restructuring of the Methods section, including the new "Software" subsection and the improved definition and justification of the vulnerability analysis at the beginning of the relevant section, has significantly enhanced readability.

Justification of Choices: The authors have adequately justified the selection of specific centrality measures (in-degree, out-degree, betweenness) over others (e.g., eigenvector) for their intervention scenarios, citing relevant literature and the specific context of directed movement networks.

Clarification of Methods: Points regarding the clarification of the vulnerability simulation process, the definition of vulnerability, and the reason for choosing 2022 for the spatial spreader analysis have been effectively resolved in the revised text.

Suggested Minor Revisions for Final Improvement:

While the manuscript is in excellent shape, a few minor points could before final publication:

Figure 9 Legend: The explanation for "Betweenness 21" is now in the main text, which is good. Ensure the final legend for Figure 9B is exceptionally clear, perhaps phrasing it as: "Impact on vulnerability when nodes are removed from the 2022 network based on the betweenness centrality values calculated from the 2021 network (Betweenness 21), compared to other strategies."

Data Availability Statement: The justification for restricted data access due to LGPD regulations is completely valid and common for this type of data. The statement is clear. It might be marginally improved by explicitly stating that the data can be accessed for research purposes upon reasonable request and approval from the Instituto Mineiro de Agropecuária (IMA), which is implied but could be slightly more direct. E.g., "Data are available from the Instituto Mineiro de Agropecuária (IMA) (contact: ima@ima.mg.gov.br) for researchers who meet the criteria for access to confidential data, as required by Brazilian law (Lei Geral de Proteção de Dados Pessoais - LGPD, Lei n° 13.709/2018)."

Conclusion:

This is a high-quality study that applies advanced network analytical techniques to a critical problem in veterinary epidemiology. The revisions have strengthened the manuscript considerably. The findings are not only academically interesting but also have direct, practical implications for designing efficient surveillance and control programs in Minas Gerais and potentially in other regions with similar livestock movement dynamics. The manuscript fully meets the publishing criteria of PLOS ONE.

I recommend acceptance after the incorporation of the very minor suggestions outlined above.

**Do you want your identity to be public for this peer review?** For information about this choice, including consent withdrawal, please see our Privacy Policy

Reviewer #2: No

---

## [Author Response · Author response to Decision Letter 2]

1 Oct 2025

Dr. Lucas D. B. Faria

Academic Editor,

PLOS ONE

Thank you for your message regarding our manuscript “Network vulnerability of cattle movement in Minas Gerais, Brazil, from 2013 to 2022, PONE-D-24-59680. We would like to thank you for the appropriate and useful suggestions regarding our manuscript. All the points made by the reviewers are consistent and improved the manuscript.

Please find the enclosed Responses to reviewer #2.

Sincerely,

Prof. Elaine Maria Seles Dorneles

Departamento de Medicina Veterinária Preventiva,

Universidade Federal de Lavras

Av. Sul UFLA - Aquenta Sol - Lavras, Minas Gerais

Post office box: 3037

Postal Code: 37200-900. Brazil.

elaine.dorneles@ufla.br

Phone: + 55 35 3829 1712

Figure 9 Legend: The explanation for "Betweenness 21" is now in the main text, which is good. Ensure the final legend for Figure 9B is exceptionally clear, perhaps phrasing it as:

"Impact on vulnerability when nodes are removed from the 2022 network based on the betweenness centrality values calculated from the 2021 network (Betweenness 21), compared to other strategies."

Answer: We appreciatte your comments and your suggestion. The change was added to the legend of Figure 9B as requested.

Data Availability Statement: The justification for restricted data access due to LGPD regulations is completely valid and common for this type of data. The statement is clear. It might be marginally improved by explicitly stating that the data can be accessed for research purposes upon reasonable request and approval from the Instituto Mineiro de Agropecuária (IMA), which is implied but could be slightly more direct. E.g., "Data are available from the Instituto Mineiro de Agropecuária (IMA) (contact: ima@ima.mg.gov.br) for researchers who meet the criteria for access to confidential data, as required by Brazilian law (Lei Geral de Proteção de Dados Pessoais - LGPD, Lei n° 13.709/2018)."

Answer: Thank you for the improvement made to our Data Availability Statement. We will include as suggested in the website.

---

## [Editor Report · Decision Letter 2]

2 Oct 2025

Network vulnerability of cattle movement in Minas Gerais, Brazil, from 2013 to 2022

PONE-D-24-59680R2

Dear Dr. Dorneles,

We’re pleased to inform you that your manuscript has been judged scientifically suitable for publication and will be formally accepted for publication once it meets all outstanding technical requirements.

Kind regards,

Lucas D. B. Faria

Academic Editor

PLOS ONE
---

## [Editor Report · Acceptance letter]

PONE-D-24-59680R2

PLOS ONE

Dear Dr. Dorneles,

I'm pleased to inform you that your manuscript has been deemed suitable for publication in PLOS ONE. Congratulations! Your manuscript is now being handed over to our production team.

Kind regards,

on behalf of

Dr. Lucas D. B. Faria

Academic Editor

PLOS ONE